# Computer versus Compensatory Calendar Training in Individuals with Mild Cognitive Impairment: Functional Impact in a Pilot Study

**DOI:** 10.3390/brainsci7090112

**Published:** 2017-09-06

**Authors:** Melanie J. Chandler, Dona E. C. Locke, Noah L. Duncan, Sherrie M. Hanna, Andrea V. Cuc, Julie A. Fields, Charlene R. Hoffman Snyder, Angela M. Lunde, Glenn E. Smith

**Affiliations:** 1Mayo Clinic Florida, 4500 San Pablo Road, Jacksonville, FL 32224, USA; 2Mayo Clinic Arizona, 13400 E. Shea Blvd, Scottsdale, AZ 85259, USA; locke.dona@mayo.edu (D.E.C.L.); cuc.andrea@mayo.edu (A.V.C.); snyder.charlene@mayo.edu (C.R.H.S.); 3Naval Hospital Jacksonville Mental Health Department, 2080 Child Street, Jacksonville, FL 32214, USA; noah.l.duncan.ctr@mail.mil; 4Mayo Clinic Minnesota, 200 1st St SW, Rochester, MN 55905, USA; hanna.sherrie@mayo.edu (S.M.H.); fields.julie@mayo.edu (J.A.F.); lunde.angela@mayo.edu (A.M.L.); 5University of Florida, Department of Clinical and Health Psychology, 1225 Center Drive, Gainesville, FL 32610, USA; glennsmith@phhp.ufl.edu

**Keywords:** activities of daily living, behavioral rehabilitation, cognitive rehabilitation, mild cognitive impairment, self-efficacy

## Abstract

This pilot study examined the functional impact of computerized versus compensatory calendar training in cognitive rehabilitation participants with mild cognitive impairment (MCI). Fifty-seven participants with amnestic MCI completed randomly assigned calendar or computer training. A standard care control group was used for comparison. Measures of adherence, memory-based activities of daily living (mADLs), and self-efficacy were completed. The calendar training group demonstrated significant improvement in mADLs compared to controls, while the computer training group did not. Calendar training may be more effective in improving mADLs than computerized intervention. However, this study highlights how behavioral trials with fewer than 30–50 participants per arm are likely underpowered, resulting in seemingly null findings.

## 1. Introduction

Studies of cognitive rehabilitation for amnestic mild cognitive impairment (MCI) have seen exponential growth over the last decade. Most of these studies have sought to determine the utility of either therapist or computer-based interventions for treating memory loss. However, there has been virtually no research into the comparative effectiveness of various cognitive rehabilitation strategies.

The focus in rehabilitation can be to improve the impaired ability itself (remediation), to learn methods to adapt to the changed ability without attempting to improve the ability itself (compensation), or both simultaneously. The computerized brain training industry has sought to largely fill the space for remediation of cognitive loss in aging and MCI. However, warnings of overstated claims of proof of cognitive improvement from the computerized brain training industry highlight the need for more research into this area [1]. In terms of cognitive impact, computerized interventions may have a positive impact on measures of attention, executive function, and memory in the limited number of MCI-based studies available [2]. In terms of transfer to everyday life, computerized studies may benefit mood (depression, anxiety, and apathy) compared to controls [3,4]. However, a recent meta-analysis showed no impact of computerized interventions on activities of daily living (ADLs) or quality of life [5]. 

Compensation for cognitive loss in MCI is usually taught through therapist-based training in which the patient is taught strategies such as mnemonic techniques or an adaptive system of taking notes or keeping calendars or lists. Studies to date suggest these types of interventions can positively impact one’s self beliefs and confidence about memory functioning and improve ADLs, but there is little evidence of benefit to mood or quality of life [5]. This prior work suggests that computerized remediation and compensation-based interventions may have different outcomes in daily life functioning for individuals with MCI. 

Our study sought to examine the impact of a computerized cognitive intervention program aimed towards older individuals (Brain Fitness (BF); Posit Science Corporation) to a calendar compensation training program (Memory Support System (MSS)) through a randomized trial. Using this particular computer program, researchers have reported that healthy participants and participants with amnestic MCI who trained on BF showed improvement in cognitive functions (overall *d* = 0.33), particularly in working memory and processing speed [6,7], but little was done to assess daily life impact. A previous study of the MSS calendar and notebook strategy found significant effect sizes on memory-based activities of daily living (mADLs) at the end of training (*d* = 1.0) and at 2- (*d* = 0.88) and 6-month follow-up (*d* = 0.56) compared to no-treatment controls [8]. 

The current study was a pilot intervention that sought to examine the relative efficacy of the calendar and computer interventions, and the effect sizes of these interventions for necessary sample sizes in future effectiveness trials was estimated. We hypothesized that the MSS compensatory training would have a greater positive impact on mADLs and memory self-efficacy than the BF program. The compensatory MSS training is meant to teach strategies to directly impact daily functioning, while the restorative BF program is meant to improve the abilities themselves without prior evidence of transfer to daily life. We report the findings of the comparative effect sizes on functional ability, cognitive functioning, and self-efficacy in individuals with amnestic MCI. 

## 2. Materials and Methods

### 2.1. Participants

Consecutive patients with single or multiple-domain, amnestic MCI from the neurology and neuropsychology clinics across 3 major medical centers (Emory University and the Minnesota and Arizona campuses of Mayo Clinic) were asked to take part in a study comparing the use of a calendar and note-taking system to a cognitive training computer program. Participants were diagnosed with either a single or multiple-domain amnestic MCI [9], as these subtypes of MCI are thought to represent individuals with a high likelihood of eventual progression to dementia [10]. All participants gave their informed consent for inclusion before they engaged in the study. The study was conducted in accordance with the Declaration of Helsinki, and the protocol was approved by the institutional review boards at Emory University and Mayo Clinic. Participants met the following inclusion criteria: (1) had a study partner (as this is a required component of the MSS training as described below); (2) obtained a Dementia Rating Scale-2 (DRS-2) [11] score of 115 or higher; (3) obtained a Functional Assessment Questionnaire [12] score lower than 6; (4) the partner had a Mini Mental Status Exam (MMSE) [13] score higher than 24; and (5) was not using nootropic medications or had been stable for at least 3 months. Participants were excluded if they had a visual or hearing impairment or reading or writing disability sufficient to interfere with training or severe depression as defined by Center for Epidemiological Studies- Depression (CES-D) [14] of 21 or higher. 

Two hundred individuals and their study partners were contacted across the 3 recruitment sites for possible enrollment in the study, and 126 declined enrollment (37% recruitment). After excluding those who did not meet eligibility criteria, 64 of the 200 individuals originally contacted were enrolled. Enrollment and retention have been further detailed in a separate publication [15]. 

### 2.2. Control Group

Comparisons were made to a standard care control group (*n* = 20) collected using the same inclusion and exclusion criteria. This sample was collected at Emory University only. This group was previously utilized in a separate calendar training study examining the efficacy of MSS training to a no-treatment control group [8]. 

### 2.3. Intervention

Participants with MCI were randomized to either computer (BF) or calendar (MSS) training using computer-generated randomization assignment. At enrollment, all participants were given the calendar and notebook and briefly instructed to “begin using the calendar to help with your memory.” At the start of their training program (7–10 days after enrollment), baseline use of the calendar was assessed, and MSS training participants began training. BF participants instead performed the cognitive exercises from the computer software and received no MSS training. All participants completed 10 h of each intervention with the therapists at the medical center. Both groups participated in an on-site education hour each day in addition to either the MSS or BF training. The education component was an adaptation and synthesis of the Savvy Caregiver psychoeducational program [16] and the Memory Club educational program [17,18]. 

#### 2.3.1. MSS Calendar Training

The MSS is made up of the following 3 sections: (1) appointments, (2) items to be done, and (3) a notes section. In the appointment and items to be done sections, participants write things that need to be done either at a specific time or in a to-do list if not due at a certain time. The notes section allows participants to record meaningful information from that day, such as an interesting discussion or thoughts on a day’s event. MSS trainers provide participants with MCI and their study partner with training sessions following a structured curriculum of orientation, modeling, practice, and homework assignments. Program partners learn the intervention alongside the trainer so that they can reinforce proper use of the MSS outside of therapy sessions. Sessions follow a training manual, which is detailed in a previous publication [8]. 

#### 2.3.2. BF Computer Training

The BF computer training group utilized software from the Posit Science Corporation that was built on the principles of brain plasticity and designed for use by older individuals. The BF training program focuses on speech reception to strengthen an individual’s auditory memory. It is an adaptive program that adjusts to the individual’s performance. The specific “Auditory Brain Training” software used in this study has 6 modules that build upon each other in complexity that require participants to perform exercises to recognize sounds, tell similar sounds apart, match or repeat sounds, remember increasingly difficult directions, and remember details from stories. 

### 2.4. Outcome Measures

Participants and their study partners completed measures of global cognitive status, mADLs, and self-efficacy at baseline, 3-month, 6-month, and 1-year follow-ups. Measures of mADLs and self-efficacy were given additionally at the end of training. 

#### 2.4.1. Global Cognitive Status

The DRS-2 and MMSE are widely used cognitive screening measures developed to track global cognitive status. Higher scores represent better cognitive functioning. These measures were used to gauge overall level of cognitive impairment and not aimed to detect cognitive change from either intervention.

#### 2.4.2. Memory-Based Activities of Daily Living

The Everyday Cognition (ECOG) [19] is an informant-based measure that assesses ability to perform higher level ADLs in the areas of memory, language, visuospatial abilities, planning, organization, and divided attention. Our study focused on the 8-item Memory subscale as a measure of mADLs. Each item was scored 1 to 4 points, with lower total scores suggesting more intact daily functioning. The informant was the program partner. 

#### 2.4.3. Self-Efficacy

The Self-Efficacy in MCI scale is a 9-item measure of self-efficacy or confidence created by adapting selected items from the Chronic Disease Self-Efficacy Scales [20] for use in MCI. Specific items focused on confidence in managing activities, tasks, and emotional distress caused by MCI; in medication management, chores, and errand ability; and in maintaining hobbies and relationships. Higher scores indicated greater self-efficacy. 

#### 2.4.4. Adherence

The Adherence Assessment measures how well an individual is using their calendar on a daily basis. The MSS Instructor examined MSS adherence for two days randomly selected from the prior week to assess use, for a maximum of 10 points:(1)brought the MSS to the appointment (1 point);(2)has at least one entry for today’s date (1 point);(3)has entries for events happening at a certain time (2 points) and any time (2 points);(4)has at least two entries for each of the two days in the journaling section (4 points).

### 2.5. Analysis

Study data were collected and managed using REDCap electronic data capture tools hosted at Mayo Clinic and Emory University [21]. Data were analyzed using SPSS software (Version 22, IBM Corp., Armonk, NY, USA). Intragroup change was analyzed using Wilcoxon rank sum or paired t-tests, as appropriate, and differences on raw or change scores between intervention groups and controls were analyzed using analysis of variance with Tukey test post-hoc comparisons. Nonparametric between group comparisons were analyzed using the Mann–Whitney test (2 sample) or Kruskal–Wallis test (3 sample). Effect sizes were calculated based on the Cohen’s d method. Sample size estimates were based on power analysis using 0.80 power with 2-tailed significance at *p* < 0.05.

## 3. Results

Randomization led to 34 participants in the MSS program and 30 participants in the BF program (*N* = 64). Seven couples withdrew prior to the start of any intervention (remaining *N* = 57). The sample size and characteristics of the participants in each group are listed in Table 1. 

DRS-2 scores were significantly different across training sites (*F*_2, 61_ = 4.1, *p* < 0.05) with higher scores at Emory University (mean (*M)* = 133.0, standard deviation (*SD*) = 7.8) than at Mayo Clinic in Scottsdale, Arizona (*M* = 127.4, *SD* = 7.5). There were no other significant differences across training sites.

### 3.1. Adherence

MSS and BF group calendar adherence scores were not significantly different at baseline (MSS = *M* = 1.9, *SD* = 2.0; BF = *M* = 1.8, *SD* = 1.4) *z* = −0.12, *p* < 0.05. The Rochester site had significantly lower adherence scores at baseline than the Emory and Scottsdale sites (Rochester *M* = 0.6, *SD* = 0.8; Emory *M* = 2.0, *SD* = 2.3; Scottsdale *M* = 2.0, *SD* = 1.4; *H(2)* = 7.4, *p* < 0.05), but there was no significant difference in adherence among sites at any other time point [training end *H(2)* = 4.0, *p =* 0.1, 3-month *H(2)* = 0.3, *p =* 0.9, 6-month *H(2)* = 0.1, *p =* 1.0, and 1-year follow-up *H(2)* = 0.1, *p =* 0.8]. The MSS training group demonstrated significantly better adherence to the MSS compared with the BF group at training end *(*MSS = *M* = 8.4, *SD* = 2.2; BF= *M* = 1.2, *SD* = 1.8; *z* = −6.2, *p* < 0.001) and at 3-month (MSS = *M* = 5.3, *SD* = 3.8; BF = *M* = 1.4, *SD* = 2.5; *z* = −3.8, *p* < 0.001), 6-month (MSS = *M* = 5.0, *SD* = 4.1; BF= *M* = 1.1, *SD* = 2.4; *z* = −3.7, *p* < 0.001), and 1-year follow-ups (MSS = *M* = 3.8, *SD* = 3.9; BF= *M* = 0.4, *SD* = 0.9; *z* = −3.5, *p* < 0.001). 

### 3.2. Impact of Cognition

There were no significant differences in the DRS-2 total score (MSS *M* = 130.2, *SD* = 8.4; BF *M* = 129.5, *SD* = 7.4) or MMSE score (MSS *M* = 25.8, *SD* = 3.2; BF *M* = 26.7, *SD* = 3.0) across groups at baseline (*p* > 0.05). DRS-2 annual follow-up scores (MSS *M* = 126.4, *SD* = 13.8; BF *M* = 127.7, *SD* = 10.1) did not differ by intervention (*F*_1, 35_ = 0.32, *p* > 0.05). 

### 3.3. Memory-Based Activities of Daily Living

#### 3.3.1. Interventions versus Controls

mADLs were significantly different among the MSS, BF, and control group by treatment end (*F*_2, 71_ = 4.4, *p* = 0.02). In post-hoc pairwise comparisons, the MSS training group had significantly improved mADLs compared to the control group (*p* = 0.01), while the BF group did not demonstrate significant improvement relative to controls (*p* > 0.05). Table 2 displays change in mADLs, effect sizes, and projected sample sizes needed to attain significance.

#### 3.3.2. MSS versus BF

There were no significant differences at the end of training between the MSS and BF groups in functional status (as rated by mADLs) in this small pilot sample (*F*_3, 51_ = 1.1, *p* > 0.05). Within-subject mADLs were significantly improved at the end of training for MSS (*t* = 2.5, *p* < 0.05), but also improved with BF (*t* = 2.3, *p* < 0.05). There were no noteworthy differences or effect sizes in mADL status between the MSS, BF, or control groups at 3-, 6-, and 12-month follow-up points (Figure 1). 

### 3.4. Self-Efficacy

#### 3.4.1. Interventions versus Controls

Overall group change in self-efficacy was not significant among the MSS, BF, or control group (*F*_2, 71_ = 2.1, *p* = 0.1). Changes in self-efficacy, effect sizes, and projected sample size needed to attain significance are displayed in Table 2. Considering within-subject change, MSS training significantly improved self-efficacy (*t*_27_ = −2.5, *p* < 0.05), while BF did not (*t*_26_ = −1.1, *p* > 0.05). The MSS group retained improvement in their self-efficacy over baseline out to 3-month follow-up (*t*_25_ = −2.3, *p* < 0.05). While not significant in this sample, the effect size for the MSS group compared to the control group at 3 months was *d* = 0.53 (power to detect significance at *n* = 53 in each group) and at 6 months was *d* = 0.26 (power to detect significance at *n* = 238). There were no meaningful differences or effect sizes between the BF and control groups at 3- or 6-month follow-ups. 

#### 3.4.2. MSS versus BF

While there were no notable differences in self-efficacy at follow-up points between the groups, there was a mild to moderate effect size (*d* = 0.37, power to detect significance at *p* < 0.05 at *n* = 118) at the end of training and a moderate effect size (*d* = 0.41, power to detect significance at *p* < 0.05 at *n* = 94) by 3 months, both in favor of the MSS training group (Figure 2).

## 4. Discussion

### 4.1. Findings

Training in the use of the MSS calendar improved mADLs relative to standard care controls, while BF computerized training did not. This is similar to what was found in a recent meta-analysis, suggesting that, while computerized programs may impact cognition and anxiety and not ADLs, compensatory-based programs working with a therapist have more everyday impact on daily life function [5]. Herein, we report effect sizes for several other comparisons that suggest that larger samples would be needed to attain significance in future studies. Based on the effect size estimates, larger sample sizes might support statistically significant differences, including the fact that MSS is superior to BF, which in turn is superior to controls for positively impacting mADLs, and memory-related self-efficacy may be improved and retained for up to 6 months for the MSS, but not a BF program. 

While initial improvement may occur in mADLs and self-efficacy, these benefits recede over time. Future studies may examine whether cognitive interventions, while not supporting continued gains over baseline, actually help with maintenance of functional ability and self-efficacy over time compared to progression or decline in controls. Additionally, future studies should look at how boosting ongoing practice of interventions may lead to more long-term maintenance of the initial gains. 

### 4.2. Challenges

#### 4.2.1. Impact of Small Sample Sizes

Repeatedly in the literature involving nonpharmacologic interventions in MCI, a small sample size is highlighted as a limitation (e.g., [22]). This is indeed a weakness of this field, and when small samples are combined with the lack of a control group (another frequent limitation), the published findings often fail to show statistical significance and may be of dubious value. The aims of this pilot study were to provide effect and sample size estimates to determine necessary sample sizes for future studies. However, sample sizes for this pilot are larger than many past efficacy studies in the field of nonpharmacologic interventions for MCI. We think it is timely to highlight necessary sample sizes for studies in this area. We would like to emphasize that studies, though well-designed, are often too small to produce significant results. To date, few studies of behavioral interventions in MCI include the 50+ patients in each arm that our results suggest are necessary to attain significance. Admittedly, the need for larger samples relates to the probability that the effect sizes produced by these interventions are modest. However, such modest effects may still be clinically meaningful [5].

It is well-known that with multiple interventions, a large number of participants will be required to show significance. We have illustrated in our study that with a small number of participants, potentially meaningful results could have been lost due to lack of significance, even with moderate to large effect sizes. Given that few studies of behavioral interventions in MCI include the 50 or more participants in each arm that our results suggest are necessary to attain significance, efforts are needed to increase the power (i.e., n), including better-funded, large-scale, multi-site trials. 

#### 4.2.2. Impact of the Heterogeneity of MCI Diagnosis

MCI is a diagnostic syndrome that is composed of multiple possible underlying pathologies [23]. As such, it is likely that the impact of the computer or calendar interventions may differ depending upon the underlying etiology of the MCI diagnosis. We do not have that data for this study. Ideally, cognitive rehabilitation interventions are geared specifically towards the condition and strengths and weaknesses of the individual [24]. Future research will be needed to define what characteristics of individuals (including underlying etiology for their MCI) relate to the best “active ingredients” of cognitive intervention(s) for a specific person. 

#### 4.2.3. Impact of Providing Education

We provided both groups education about MCI in order to help control treatment diffusion in this domain as many individuals with MCI are seeking information through seminars, web searches, and local community events. Therefore, it is not possible to know how much of the shared improvement between the MSS and BF training was related to the education received by all participants in this study. 

#### 4.2.4. Impact of Subjective Report Measures

Our measures of functional change and self-efficacy are subjective reports of performance, with functional ability reported by the care partner and sense of self-efficacy reported by the patient. While the informant report on the ECOG is a proxy for mADLs, these reports may not capture exact functional ability. Both self and informant reports are at risk in outcome studies of bias due to a possible desire to please the evaluator. However, informant and self-reported measures of activities of daily living remain the primary tools for the detection of change in functional ability in MCI in the field, and can be more sensitive than cognitive testing towards predicting progression of disease [25]. As each treatment arm in the current study completed these measures pre- and post-intervention, we attempted to minimize any positive bias in outcome reporting that would have been more prevalent in examining a single intervention arm. 

### 4.3. Further Clinical Implications

In past studies, we have demonstrated that use of a calendar is dependent upon having training in the calendar [8]. Individuals with MCI do not use a calendar (even when given the calendar to use) with only a recommendation. The present study helps further support these findings, as individuals in the BF group were given a calendar, encouraged to use it, and knew that the other group (whom they joined for education classes) were trained to do so. While most individuals in the BF training group brought their calendars daily along with their general study information packets, they failed to use them. Thus, recommending a calendar is not sufficient for an individual with MCI, and as providers, we need to keep this under consideration with our patients.

## 5. Conclusions

Calendar training may be more effective in improving mADLs and sense of self-efficacy than computerized intervention. However, this pilot study highlights how behavioral trials with fewer than 30–50 participants per arm are likely underpowered. As a result, modest, but still potentially beneficial interventions may be dismissed as null findings. 

## Figures and Tables

**Figure 1 brainsci-07-00112-f001:**
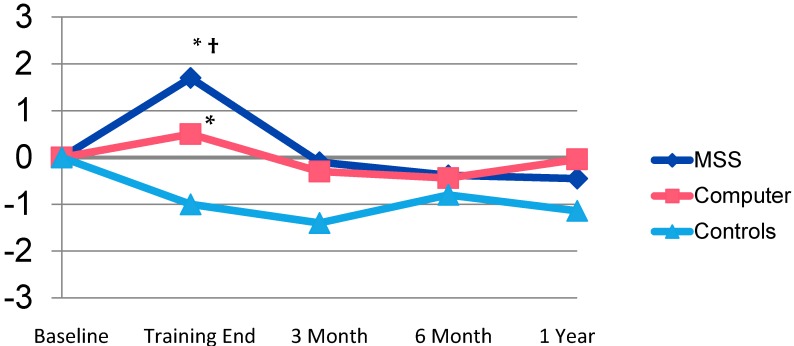
Change in Memory Activities of Daily Living Over Time. * *p* < 0.05 within-subject change. † *p* = 0.01 between-subject change for Memory Support System (MSS) compared to controls.

**Figure 2 brainsci-07-00112-f002:**
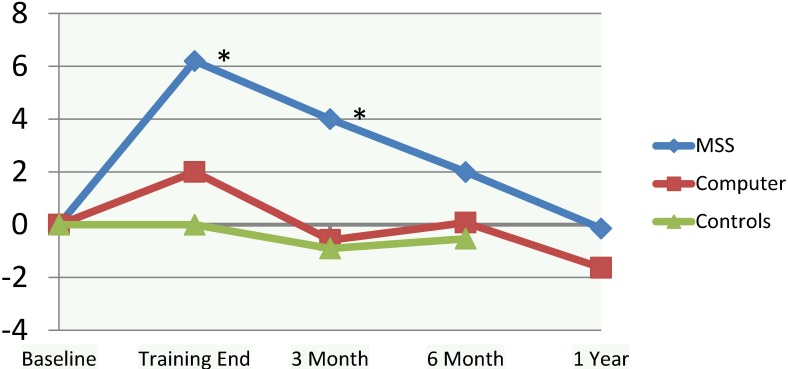
Change in Self Efficacy Over Time. ** p* < 0.05 within-subject change.

**Table 1 brainsci-07-00112-t001:** Patient Characteristics.

Characteristic	Patient Group		
	MSS (*n* = 30)	BF (*n* = 27)	Controls (*n* = 20)
Age, Mean (SD)	76.2 (7.0)	77.4 (7.2)	72.3 (7.9)
Education, Mean (SD)	16.0 (2.4)	16.2 (2.6)	16.4 (2.7)
Sex, male, %	50.0	73.3	55.0
White, %	91.1	90.0	85.0
Spouse as program partner, %	91.2	90.0	75.0
AChE use, %	54.5	40.0	70.0

AChE: acetylcholinesterase inhibitor; BF: brain fitness; MSS: memory support system; SD: standard deviation. All differences between groups were not significant.

**Table 2 brainsci-07-00112-t002:** Change in Memory Activities of Daily Living and Self-Efficacy at the End of Training.

	Mean Change (SD)	Compared to Control, Cohen’s *d* (Significant at)	Calendar vs Computer, Cohen’s *d* (Significant at)
Activities of daily living			
MSS	−1.7 (3.3) ^a^	*d* = 0.75 ^b^	*d* = 0.39 (*n* = 105)
BF	−0.5 (2.2) ^a^	*d* = 0.54 (*n* = 55)	
Controls	1.0 (3.3)		
Self-efficacy			
MSS	6.2 (12.9) ^a^	*d* = 0.52 (*n* = 58)	*d* = 0.37 (*n* = 118)
BF	2.0 (9.3)	*d* = 0.22 (*n* = 323)	
Controls	0.0 (9.2)		

BF: brain fitness; MSS: memory support system; SD: standard deviation. Negative scores represent improvement in activities of daily living, whereas positive scores denote improvement for self-efficacy. For this small pilot study, when the findings were non-significant, per group sample size predictions to achieve significance are provided. ^a^
*p* < 0.05 for within-subject change ^b^
*p* = 0.01 compared to control group.

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
