# Peer review of "Computer versus Compensatory Calendar Training in Individuals with Mild Cognitive Impairment: Functional Impact in a Pilot Study"

_brainsci, 2017, doi:10.3390/brainsci7090112_

Round 1
Reviewer 1 Report
This manuscript is well written and succinctly presents information regarding a significant issue in the field of cognitive rehabilitation for amnestic Mild Cognitive Impairment (MCI). The authors examined the functional impact of computerized versus compensatory calendar training in participants with MCI. Focus on functional impact, as measured by informants’ ratings of the participants’ memory, and self-efficacy facilitated assessment of “real world” impact of the interventions. The authors also highlight key issues of trial power and design issues that exist in the current MCI literature. This information has the potential to greatly enhance future research focused on MCI interventions. I recommend publication of this manuscript given the rigor of how this study was conducted and the novelty of the data reported, which has the potential to greatly enhance future research focused on MCI interventions.
The following recommendations and suggestions are offered to improve the manuscript for publication.
1. The abstract mentions adherence data was collected but data do not appear elsewhere in the manuscript. Brief discussion of adherence of the final sample, and any site-specific differences, is recommended. Data regarding baseline use of the calendar at the start of the training program would also be good to include.
2. Data regarding percentages of single vs. multiple-domain amnestic MCI, and which domains are included in the multiple-domain participants, would be useful to include to further characterize this sample and for future comparison purposes.
3. The intervention is noted to include instruction to use a calendar for all participants as well as an education hour each day. Did participants complete all the study sessions on site or at home? Was the time engaged in the intervention equivalent for the MSS and BF groups? What was the amount of time engaged in the interventions? These would be helpful to include, so that readers can compare the two interventions within this manuscript and then reference prior studies as noted in the manuscript if desired for more details.
4. I suspect that the authors instructed all participants to use the calendar for face-validity reasons as well as the treatment diffusion issue they discussed in 4.2.2, but inclusion of the rationale of why the calendar was instructed to all participants would be helpful to include.
5. I suggest that the authors consider changing “measures of cognition” to “measures of global cognitive status” in the Outcome Measures paragraph 2.4 so that it is clear that there are no other measures of cognition besides these global screeners as they clearly explain in the Cognition paragraph 2.4.1.
6. In the 3.2.1 Inventions Versus Controls paragraph, statement of whether the mADLs differed at 3, 6, and 12 month follow up would be good to include.
7. In the 3.2.2 paragraph, I had to re-read the manuscript to clarify that “functional status” referred to the mADLs data. It may be good to include a brief reference to mADLs, such as “in functional status, as rated by mADLs, ….” So that this is clear to readers.
8. The discussion is very well developed and clearly details limitations as well as critical issues for the field. One sentence that starts with “to date, few studies of behavioral interventions…” is repeated twice on line 228 and 235. Given how close these sentences are, editing is recommended. The second reference to the issue could be stated something like, “Given that few studies include 50 or more…, efforts are needed to increase….”
Author Response
We thank Reviewer 1 for the positive statements about the importance of our work and the general writing of the paper. We also thank the Reviewer for their constructive criticism to improve our manuscript, and we have addressed their concerns as follows:
We have added information about the adherence data to the results and discussion sections.
Unfortunately, the delineation of whether an individual was single domain or multiple domain was not coded in all cases for us to report these percentages accurately. While all cases were single or multidomain amnestic MCI, 64% of the sample was not coded in more detail. A remaining 20% were coded single domain and 16% were coded multiple domain.
We added information about the equivalent times spent in each intervention on site during the study to the methods section.
We added our rationale for giving all participants a calendar in the discussion section “further clinical implications.”
This change has been made.
We clarified in the next section 3.2.2 that data for the 3, 6, and 12 month follow ups is provided in the figure for the control group as well as the intervention groups.
This clarification was made.
The rewording of the repetition was made as suggested.
Reviewer 2 Report
The paper is of great interest for scientific readers because it reports on potential interventions to prevent cognitive decline and functional impairment in old age. I would like to make some minor suggestions that could help to improve the manuscript,
Line 76-77, how exactly did you determine whether somebody has a "high likelihood of eventual progression to dementia"?
Could you mention shortly why there is a "study partner" and who the study partner is (in terms of demographic characteristics)?
Lines 115-121, I am interested in more details on the BF Computer Training. What exactly was the tasks that the participants had to do? How was the training developed? Has it been used before?
It would be helpful to have a timeline of the follow-ups. What is the time interval between them? When you report results, it would be great to know to what follow-up time point you are referring to.
In addition to the mean change, I would like to read about the mean values at baseline and follow-up. Is it possible to add this information?
Lines 202-203, could you add p-values?
In the discussion, I think it is important to mention that MSS can temporarily improve ADLs and self-efficacy, but the effect decays when the training has ended (that´s what the results tell me). Maybe the authors can use the conclusion to discuss the practical implications of such an intervention in practice and point out what type of study should be conducted next?
It could be worth discussing that the MCI group is a mixed group with different types of pathology. For some of them, it is not sure whether they will get dementia. The results may differ with respect to pathology.
Author Response
We thank Reviewer 2 for the positive statements about the importance of our work. We also thank the Reviewer for their constructive criticism to improve our manuscript, and we have addressed their concerns as follows:
Line 76-77 : We had written this poorly. We clarified that amnestic MCI is thought to represent a diagnostic group with a “high likelihood of eventual progression” rather than that we had identified people based on any additional criteria.
We added a description of the study partner to the Methods section and basic demographic (i.e., relationship to the patient) is found in Table 1.
We clarified in the Methods section that the BF Computer Training was from Posit Science and described the 6 modules included in the software package and the theoretical basis of its development. Information on its previous use is provided in the Introduction section.
We had added clarification that after training end, follow ups occurred at 3, 6, and 12 months post intervention as needed throughout the text.
While we do have data on the raw scores rather than just change scores, we have found that in the past when we have provided both sets of scores in this and related research, the readers or our presentation audience are confused by this. Namely, there is anticipation that the difference they see between the mean raw score at time 1 and time 2 should equal the change score we provide between time 1 and time 2. This is not the case, as the overall mean values at each time point do not equate to the mean individual change for subjects on that variable. As it is the change in variable value we find most salient to the efficacy of the intervention on a variable, we have purposefully not provided the raw scores to not get this anticipated confusion amongst the readers/audience. We respectfully hope the Reviewer finds this explanation satisfactory.
Significance of p < .05 added.
A paragraph was added to the Findings section of the Discussion about the initial improvements and then decay of benefit in mADLs and self-efficacy, and how this may be addressed in future research.
We added a section entitled “Impact of the Heterogeneity of MCI Diagnosis” to the Discussion section to address these important points about the mixed pathology in a group with MCI.